# Malnutrition as a Strong Predictor of the Onset of Sarcopenia

**DOI:** 10.3390/nu11122883

**Published:** 2019-11-27

**Authors:** Charlotte Beaudart, Dolores Sanchez-Rodriguez, Médéa Locquet, Jean-Yves Reginster, Laetitia Lengelé, Olivier Bruyère

**Affiliations:** 1WHO Collaborating Centre for Public Health Aspects of Musculoskeletal Health and Aging, Division of Public Health, Epidemiology and Health Economics, University of Liège, CHU—Sart Tilman, Quartier Hôpital, Avenue Hippocrate 13 (Bât. B23), 4000 Liège, Belgium; dolores.sanchez@uliege.be (D.S.-R.); medea.locquet@uliege.be (M.L.); jyr.ch@bluewin.ch (J.-Y.R.); llengele@uliege.be (L.L.); olivier.bruyere@uliege.be (O.B.); 2Geriatrics Department, Parc de Salut Mar Rehabilitation Research Group, Hospital del Mar Medical Research Institute (IMIM), Universitat Pompeu Fabra, 08002 Barcelona, Spain; 3Chair for Biomarkers of Chronic Diseases, Biochemistry Department, College of Science, King Saud University, Riyadh 11451, Saudi Arabia

**Keywords:** sarcopenia, EWGSOP2, malnutrition, GLIM, SarcoPhAge

## Abstract

This study aims to explore the association between malnutrition diagnosed according to both the Global Leadership Initiative of Malnutrition (GLIM) and the European Society of Clinical Nutrition and Metabolism (ESPEN) criteria and the onset of sarcopenia/severe sarcopenia, diagnosed according to the European Working Group on Sarcopenia in Older People 2 (EWGSOP2) criterion, in the sarcopenia and physical impairment with advancing age (SarcoPhAge) cohort during a four-year follow-up. Adjusted Cox-regression and Kaplan-Meier curves were performed. Among the 534 community-dwelling participants recruited in the SarcoPhAge study, 510 were free from sarcopenia at baseline, of whom 336 had complete data (186 women and 150 men, mean age of 72.5 ± 5.8 years) to apply the GLIM and ESPEN criteria. A significantly higher risk of developing sarcopenia/severe sarcopenia during the four-year follow-up based on the GLIM [sarcopenia: Adjusted hazard ratio (HR) = 3.23 (95% confidence interval (CI) 1.73–6.05); severe sarcopenia: Adjusted HR = 2.87 (95% CI 1.25–6.56)] and ESPEN [sarcopenia: Adjusted HR = 4.28 (95% CI 1.86–9.86); severe sarcopenia: Adjusted HR = 3.86 (95% CI 1.29–11.54)] criteria was observed. Kaplan-Meier curves confirmed this relationship (log rank *p* < 0.001 for all). These results highlighted the importance of malnutrition since it has been shown to be associated with an approximately fourfold higher risk of developing sarcopenia/severe sarcopenia during a four-year follow-up.

## 1. Introduction

Malnutrition is a major cause of adverse health consequences, such as impaired physical function [1], hospitalization [2], and mortality [3,4] in older people. One of the most prominent features of malnutrition is that it is a reversible disease, and a wide variety of effective therapeutic approaches are available and adaptable to the different etiologies and patient requirements [5,6].

In 2016, the World Health Organization launched the World Report on Ageing and Health, an action plan to promote initiatives towards a better ageing process [7]. The European Society of Clinical Nutrition and Metabolism (ESPEN) followed the WHO’s strategy, revisited the concepts of malnutrition and nutrition-related diseases. ESPEN developed malnutrition criteria [8] and guidelines on the definition and terminology of clinical nutrition [9] which unified the terminology to be used in malnutrition and nutrition-related diseases, i.e., sarcopenia, frailty, cachexia/disease-related malnutrition, and starvation-related underweight [8], and organized them as a conceptual tree of nutritional disorders [9]. The ESPEN approach is a two-tier process: In the first step, patients are identified as being at risk of malnutrition by any validated screening tool; in the second step, malnutrition is defined by a combination of weight loss, low body mass index, and low muscle mass [8].

The efforts by the WHO and ESPEN to shed light on malnutrition and nutrition-related diseases have been followed by the largest societies of clinical nutrition and metabolism. In the malnutrition field, the Global Leadership Initiative on Malnutrition (GLIM) [10] launched the GLIM criteria, the first international definition of malnutrition [11]; in the sarcopenia field, the European Working Group on Sarcopenia in Older People has published the revised European consensus on definition and diagnosis (EWGSOP2) [12], which updates the most widely acknowledged previous definition.

The GLIM criteria are a three-step approach, first, patients are identified by any validated screening tool, and second, they are diagnosed for presence of, at least, one phenotypic (weight loss, low body mass index, and low muscle mass) and one etiologic criteria (reduced food intake or assimilation or disease burden and inflammation). A third step is severity grading, which is based on the phenotypic criteria. The EWGSOP2 consensus follows this same three-step approach, first, screening by SARC-F questionnaire, and second, patients are diagnosed in presence of low muscle strength and low muscle mass. The third step is severity grading, based on the impairment of physical performance. The GLIM and EWGSOP2 criteria are harmonized definitions that share muscle mass as a criterion to enhance the comparability of studies [13], and sarcopenia has loss of muscle function as its most highlighted differential feature [14].

Nutritional intake is one of the most important modulators in human health, and an inadequate balance between intake and expenditure is the main cause of malnutrition [14] and nutrition-related diseases [13,15]. The association between a poor balanced diet with reduced micro and macronutrients and the presence of sarcopenia at baseline in community-dwelling older people has been recently described by our research group [16]. Likewise, malnutrition must be decisive for the onset of sarcopenia. However, the prospective associations between the two diseases remain unknown, and the incidence of sarcopenia in longitudinal studies is truly unexplored.

Our research group has followed the call to action launched by the GLIM and ESPEN to shed light on the overlap between malnutrition and nutrition-related diseases [10]. Our objective is to assess the relationship between baseline malnutrition according to the GLIM and the ESPEN criteria and the incidence of sarcopenia and severe sarcopenia in the sarcopenia and physical impairment with advancing age (SarcoPhAge) cohort during a four-year follow-up. 

## 2. Materials and Methods

This was a prospective, descriptive study cohort. The Strengthening the Reporting of Observational Studies in Epidemiology (STROBE) statement was followed [17].

### 2.1. Population

Participants from the SarcoPhAge cohort were included in this study. The protocol of the SarcoPhAge study has been detailed elsewhere [18]. Briefly, the SarcoPhAge cohort is a Belgian population-based cohort developed in Liège (Belgium) involving 534 community-dwelling participants 65 years of age and older. Participants were recruited in 2013 from press advertisements and general, geriatric, osteoporosis, rehabilitation, and rheumatology outpatient clinics and were followed up each year (T0/baseline and T1, T2, T3, T4, corresponding, respectively, to one year, two years, three years, and four years of follow-up) with a clinical examination and questionnaires. No specific exclusion criteria related to health or demographic characteristics were applied, except for the exclusion of individuals with an amputated limb or with a BMI above 50 kg/m^2^. Written informed consent was provided by participants, and the study was approved by the ethics committee of our institution (reference 2012/277).

The outcome measure was the incidence of sarcopenia/severe sarcopenia measured annually, i.e., the number of new cases each year, which were cumulated.

### 2.2. Data Collection

#### 2.2.1. Malnutrition Diagnosis

Malnutrition was diagnosed at baseline (T0) according to the two most updated definitions of malnutrition: The ESPEN [8] and the GLIM criteria [19].

The ESPEN criteria [8] propose two ways to diagnose malnutrition. Alternative one: Body mass index (BMI) <18.5 kg/m^2^. Alternative two: Unintentional weight loss combined with a low age-related BMI (<20 kg/m^2^ in <70 years or <22 kg/m^2^ in ≥70 years) or low fat-free mass index (FFMI) (<17 kg/m^2^ in men and <15 kg/m^2^ in women).

The GLIM criteria [19] require at least one phenotypic criterion and one etiological criterion, as summarized in Table 1.

#### 2.2.2. Sarcopenia Diagnosis

For sarcopenia diagnosis, we applied the latest criteria published by the EWGSOP, the EWGSOP2 criteria [12]. A complete diagnosis of sarcopenia was performed at baseline and at each time of follow-up (T1, T2, T3, and T4). The incidence of sarcopenia was thereby measured.

Confirmed sarcopenia was considered when participants presented both of the following:(1)Low muscle strength (expressed in kg). Muscle strength was measured with a handgrip hand-held dynamometer (Saehan Corporation, MSD Europe Bvba, Brussels, Belgium) calibrated at the beginning of the study and at each year of follow-up for 10, 40, and 90 kg. We followed standardized procedures by asking participants to squeeze as hard as possible three times per hand. The highest value of the six measurements was considered in our analyses (Southampton protocol) [24]. Low muscle strength is defined as <27 kg in men and <16 kg in women [12].(2)Low muscle mass. Muscle mass was measured with a dual X-ray absorptiometer (Hologic Discovery A, USA), which was calibrated daily. Fat-free mass and appendicular lean mass, obtained from whole-body DXA scans, were divided by height squared (kg/m^2^) to obtain the fat-free mass index and appendicular lean mass index (ALMI) values, respectively. A low muscle mass is defined as FFMI <17 kg/m^2^ in men and <15 kg/m^2^ in women or ALMI <7 kg/m^2^ in men and <5.5 kg/m^2^ in women.

Moreover, if a person also presented low physical performance (measured by the Short Physical Performance Battery test [24] through the assessment of balance, walking speed, and the chair stand test with ≤8 points as the threshold, or measured by a 4 m gait speed test with <0.8 m/s as the threshold), that person was considered to have “severe sarcopenia”. Physical performance was measured following the standardized assessment recommended by the European Society for Clinical and Economic Aspects of Osteoporosis, Osteoarthritis, and Musculoskeletal Diseases (ESCEO) [25].

### 2.3. Covariates

During the annual follow-up of the SarcoPhAge participants, a large number of covariates were also collected. Among these variables, we recorded the number of comorbidities that the participants were affected by and the number of drugs consumed, self-reported by each individual; the cognitive status, assessed by the mini-mental state examination (MMSE) [26]; the participants functional limitations in instrumental activities of daily living (IADLs), measured with the Lawton scale [27]; as well as the physical activity level, self-reported as the time spent in different physical activities in the past seven days based on the Minnesota Leisure Time Activity Questionnaire below, an established cut-off based on sex [28].

### 2.4. Statistical Analysis

The normality of the variables was checked by examining the histogram, the quantile–quantile plot, the Shapiro–Wilk test, and the difference between the mean and the median values. Quantitative variables following a Gaussian distribution were expressed as the mean ± standard deviation; quantitative variables not following a Gaussian distribution were expressed as the median (25th percentile–75th percentile). Qualitative variables were described by absolute and relative (%) frequencies.

First, the number of participants diagnosed with sarcopenia according to the EWGSOP2 criteria was measured. We excluded those participants from our database to allow us to measure the incidence of sarcopenia from a sample of participants free from the disease.

Second, the number of participants diagnosed with malnutrition according to either GLIM or ESPEN criteria was measured. To assess agreement between the criteria, we reported the Cohen kappa coefficient and its 95% confidence interval (CI) (overall concordance rate). Participants’ baseline characteristics were compared between those diagnosed with malnutrition with either the ESPEN criteria or the GLIM criteria and those not diagnosed with malnutrition through a Student’s t test for quantitative variables that followed a normal distribution, the Mann-Whitney U test for quantitative variables that did not follow a normal distribution, and a χ^2^ test for qualitative or binary variables.

Third, the incidence of sarcopenia and severe sarcopenia was measured each year, i.e., number of new cases each year, which were cumulated. For both the ESPEN and GLIM definitions of malnutrition, the incidence of sarcopenia/severe sarcopenia was measured in each group (malnourished versus well-nourished) and compared using a χ^2^ test. Since survival data were available (months of follow-up), we also applied the Cox proportional hazards model, giving the hazard ratio (HR) and 95% CI to measure the risk of developing sarcopenia/severe sarcopenia across four years of follow-up according to the baseline nutritional status. A crude HR as well as an adjusted HR were calculated, taking into account covariates that could potentially impact on muscle health and nutritional status: Sex, age, the number of concomitant diseases, the number of drugs, cognitive status, and the level of physical activity [18,29,30,31,32]. To avoid over adjustment with sarcopenia, we chose not to include BMI as a covariate. Survival curves were evaluated using the Kaplan-Meier method to explore the influence of malnutrition on the risk of developing sarcopenia/severe sarcopenia. Log-rank tests were performed.

Data were processed using the SPSS Statistics 24 (IBM Corporation, Armonk, NY, USA) software package. All results were considered statistically significant at the 5% critical level. 

## 3. Results

### 3.1. Population and Diagnosis of Malnutrition

Of the 534 older adults included in the SarcoPhAge study, 510 were free from sarcopenia, as diagnosed with the EWGSOP2 definition, and they constituted our baseline population. Of those 510 participants, 416 were interviewed throughout the four-year follow-up period (94 individuals were either lost to follow-up, refused to participate, were unable to continue the study, or were dead) (Figure 1).

Finally, only 336 participants had the blood samples available that were needed to assess inflammation for the diagnosis of malnutrition according to the GLIM criteria. Our final study population at baseline was therefore composed of 336 participants (Figure 1), 55.4% women, aged 72.5 ± 5.8 years with a mean of four concomitant diseases per participant and a mean of 5.6 daily consumed drugs per participant. The population was free from cognitive disorders, with an MMSE mean score of 28.3 ± 1.8 points out of 30 (Table 2).

Malnutrition, according to the ESPEN criteria, was present in 19 individuals (5.65%) and, according to the GLIM criteria, was present in 59 individuals (17.6%). Agreement between both definitions was low, with a Cohen kappa coefficient of 0.30 (95% CI 0.16–0.43). Once diagnosed with either the ESPEN or the GLIM criteria, malnourished participants presented a significantly lower BMI, a lower amount of lean mass, and ALMI as well as a lower muscle strength (the latest being applicable for male participants only) than well-nourished individuals (all *p*-values < 0.05). Participants diagnosed with malnutrition using the GLIM criteria also presented a higher number of concomitant diseases than well-nourished participants (*p* = 0.005). No other significant differences between groups were observed for the collected characteristics of the population.

### 3.2. Incidence of Sarcopenia

From baseline to four years of follow-up, 46 new cases of sarcopenia (13.7%) and 26 new cases of severe sarcopenia (7.74%) were reported. In participants diagnosed with malnutrition at baseline, regardless of the criteria used for the diagnosis, the incidence of sarcopenia was significantly higher than that in well-nourished individuals (Table 3). Among the 19 individuals with malnutrition according to the ESPEN criteria, seven (36.8%) developed sarcopenia throughout the four-year follow-up period, compared to 12.3% in the group of well-nourished participants. After adjusting for age, sex, the number of concomitant diseases, the number of drugs consumed, cognitive status, and the level of physical activity, an HR of 4.28 (95% CI 1.86–9.86) was found, revealing that malnourished participants, according to the ESPEN criteria, had a 4.28-fold higher risk of becoming sarcopenic within four years. The HR for severe sarcopenia was 3.86 (95% CI 1.29–11.54). Among the 59 individuals with malnutrition according to the GLIM criteria, 16 (27.1%) developed sarcopenia, compared to 10.8% among well-nourished participants (*p* = 0.001). An adjusted HR of 3.23 (95% CI 1.73–6.05) was found for the association with sarcopenia and of 2.87 (95% CI 1.25–6.56) for the association with severe sarcopenia. 

Figure 2 depicts the analysis of the four-year incidence of sarcopenia and severe sarcopenia for participants with baseline malnutrition according to the GLIM and ESPEN criteria. Still, confirmed with Kaplan-Meier analyses, a significant impact of malnutrition on the onset of sarcopenia and severe sarcopenia was found, regardless of the definition used for malnutrition (log rank *p* < 0.001 for all Kaplan-Meier curves) (Figure 2A–D).

## 4. Discussion

Our study had the objective to explore the association between malnutrition diagnosed according to both GLIM and ESPEN criteria and the onset of sarcopenia and severe sarcopenia. Using the SarcoPhAge study population, we found an approximately fourfold higher risk of developing sarcopenia in patients meeting the ESPEN criteria and a threefold higher risk in patients meeting the GLIM criteria during a four-year follow-up, and this association was independent of the number of concomitant diseases, the number of drugs, cognitive status, and the level of physical activity at baseline.

The prevalence of malnutrition at baseline in our sample cohort of community-dwelling older people by using the GLIM criteria (17.6%) was three times higher than the prevalence obtained by applying the ESPEN criteria (5.65%), a prevalence that might serve as a reference in community-dwelling older populations in the absence of previous reference values. This is consistent with previous studies: A prevalence of malnutrition according to the ESPEN criteria of 7.3% and with a higher mortality risk (adjusted HR = 4.4 (95% CI: 1.7–11.3)) during a seven-year follow-up period in the EPIDOS-Toulouse study was found [4]. A study conducted in advanced cancer patients assessed the prevalence of malnutrition according to the GLIM criteria (by using FFMI to measure reduced muscle mass) and the association with mortality; a prevalence of 72.2% and an odds ratio of 1.87 (95% CI 1.01–3.48, *p* = 0.047) for six-month mortality were found [33]. In hospitalized patients with hematological malignancy, the prevalence of malnutrition according to the GLIM criteria was 25.8%, and the one-year mortality risk HR was 2.39 (1.36–4.20, *p* =  0.002) [34]. In outpatients with liver disease under evaluation for liver transplantation, the prevalence of malnutrition according to the GLIM criteria was 25% [35]. The high prevalence and the strong relationship with clinical adverse consequences observed for the GLIM criteria might be due to the criteria that form the definition, as all of them are important predictors of poor prognosis and unintentional weight loss [36].

Regarding the incidence of sarcopenia, 13.7% of the participants in the SarcoPhAge cohort study developed sarcopenia during the four-year follow-up period, and 7.7% developed severe sarcopenia, with a significantly higher incidence in malnourished individuals. In our study, the ESPEN criteria seemed to be slightly more related to the incidence of sarcopenia than the new GLIM criteria. The links between malnutrition and sarcopenia have already been explored in several cross-sectional studies: In older patients with advanced chronic disease (Stages 3b–5) by using the malnutrition inflammation score (MIS) for the diagnosis of malnutrition and the EWGSOP2 for the diagnosis of sarcopenia [37], in older patients with chronic pulmonary disease by using the ESPEN [38] and the EWGSOP2 [38] criteria, and in older people discharged from post-acute care by using the ESPEN and the EWGSOP criteria [39]. However, the cross-sectional study design did not allow the authors to establish cause-effect relationships. Unfortunately, there are few longitudinal studies on malnutrition and sarcopenia that allow us to compare our findings. The multicenter prospective Gruppo Lavoro Italiano Sarcopenia—Trattamento e Nutrizione (GLISTEN) study provided an incidence of sarcopenia in hospitalized older people in acute care during a hospital stay, with 15% of new cases of sarcopenia during hospitalization (10 days) [40]. The GLISTEN study did not assess the potential impact of malnutrition on the onset of sarcopenia but highlighted that a higher incidence seemed related to ADL disability and the length of bed rest. Moreover, they highlighted a decreased probability of developing sarcopenia in patients with a higher BMI and higher skeletal muscle index [40]. Our results indicated that malnutrition seems to be one of the risk factors for sarcopenia, but the onset of sarcopenia is more than likely multifactorial, with other factors, such as sedentary lifestyle, inflammatory biomarkers, and poor balanced nutrition, that should be investigated in longitudinal cohort studies for their role as risk factors.

The role of malnutrition in the onset of sarcopenia identified in our study could partially be explained by the fact that some nutritional factors, such as protein, vitamin D/calcium, and the acid–base balance of the diet, play an important role in maintaining muscle mass [41] and, consequently, muscle strength and physical performance. This overlap between malnutrition and sarcopenia is observed throughout their management. Indeed, the treatment of sarcopenia is based on a combined intervention of nutritional therapies and resistance training, with a higher influence of the second one, particularly high-intensity resistance training (i.e., 80% 1-Repetition Maximum) to gain maximal strength or low-intensity resistance training (≤50% 1 RM) to induce strength gains [42]. The therapeutic interventions for malnutrition have been recently revisited and updated in the ESPEN guidelines of clinical nutrition and hydration in geriatrics [6] and are mostly based on food fortification [6], which is more feasible to be administered in older people than the recommended management of sarcopenia [42,43]. Our findings might be of interest in the development of early therapeutic interventions targeted at individuals who meet malnutrition criteria but are free from sarcopenia at baseline [43,44] in relation to the concept of “impactability”, a term used in public health management strategies “to identify patients who are most likely to benefit from a therapeutic intervention” [45]. The placebo-controlled design of those eventual trials might present ethical issues, as malnutrition should be treated once diagnosed [6]. Very recently, considering the closer overlap between sarcopenia and malnutrition, a new clinical syndrome proposal was made, those of malnutrition sarcopenia syndrome (MSS) [46].

### Strengths and Limitations

The assessment of sarcopenia and malnutrition, according to the most updated definitions, should be highlighted as a novelty and strength of our study. Indeed, a recent review of nutrition and sarcopenia screening tools highlighted the lack of standardization and use of validated and highly recognized tools to diagnose both malnutrition and sarcopenia [46]. In our study, we did not apply the screening part of both definitions. Screening could be very useful in clinic research to avoid a full diagnosis in participants for whom the screening revealed a low risk of malnutrition. However, since we had a dataset with all participants, we had sufficient data to directly apply the diagnostic criteria for our whole population.

A limitation of the study is that we measured only the causal relationship between malnutrition and sarcopenia in the sense of malnutrition being a risk factor for sarcopenia. Assessing the incidence of malnutrition in individuals with sarcopenia at baseline during the four-year longitudinal follow-up is an interesting topic for further research to better interpret the knowledge derived from our current findings and to complete the current knowledge about the pathophysiology of sarcopenia throughout the lifespan [13]. In the same vein, the dynamic aspect of malnutrition could also be considered with an analysis that does not only focus on the baseline prevalence of malnutrition but that also takes into account the incidence or the new-onset of malnutrition as a risk factor for sarcopenia. Unfortunately, in the present study, we did not have the sufficient materials to measure the malnutrition according to both criteria at each time of data collection. There was also a potential selection bias linked to cohort studies, as volunteers might present better health status than the general population. Our results are therefore non-representative of the population of interest and could not be generalized to other populations. Because our people with sarcopenia were probably in better health than the “true” population of people with sarcopenia, the association measured in our study could have been somewhat underestimated. Finally, the fact that we used an existing dataset for these analyses implies, first, that we could have missed some important covariates that could explain the incidence of sarcopenia outside of malnutrition and, second, that no power size has been calculated. However, even with a low prevalence of malnutrition in our sample (*n* = 19 for ESPEN, *n* = 59 for GLIM), a significant relationship with the incidence of sarcopenia has already been found. We can then assume that a larger sample size and a larger number of malnourished individuals will result in an even more important difference.

## 5. Conclusions

In conclusion, malnutrition was found to be a strong predictor of sarcopenia and severe sarcopenia during a four-year follow-up. Our research suggests that both the ESPEN and GLIM criteria might be early indicators to identify those individuals free from the disease that might develop sarcopenia in the upcoming years and to shed light on the physiopathology of sarcopenia throughout the lifespan.

## Figures and Tables

**Figure 1 nutrients-11-02883-f001:**
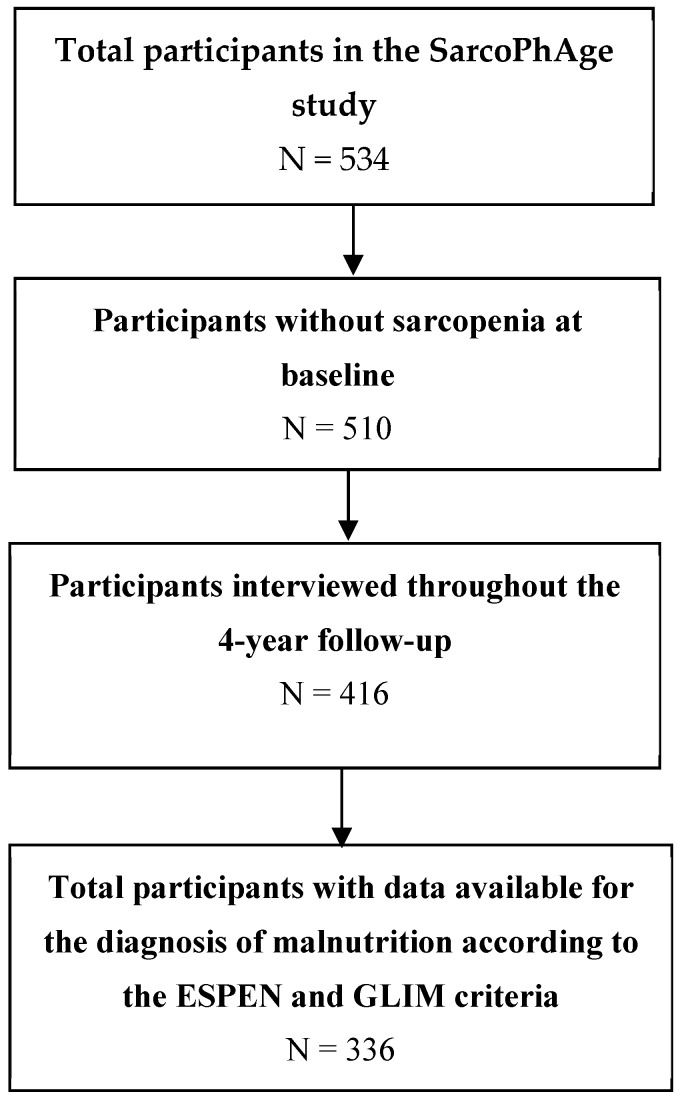
Flow chart of the study. European Society of Clinical Nutrition and Metabolism (ESPEN) and sarcopenia and physical impairment with advancing age (SarcoPhAge).

**Figure 2 nutrients-11-02883-f002:**
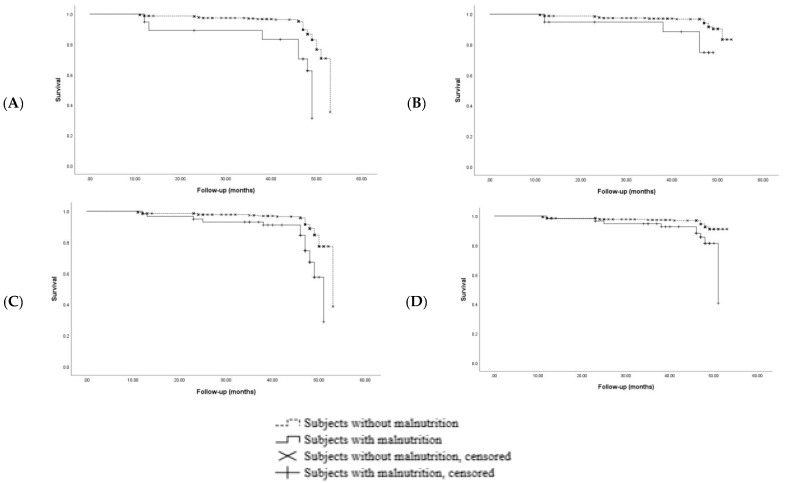
(**A**) Incidence of sarcopenia in participants with or without malnutrition according to the ESPEN criteria; (**B**) incidence of severe sarcopenia in participants with or without malnutrition according to the ESPEN criteria; (**C**) incidence of sarcopenia in participants with or without malnutrition according to the GLIM criteria; and (**D**) incidence of severe sarcopenia in participants with or without malnutrition according to the GLIM criteria.

**Table 1 nutrients-11-02883-t001:** Summary of phenotypic and etiological criteria of the Global Leadership Initiative on Malnutrition (GLIM) definition.

Phenotypic	
Weight loss	A weight loss >4.5 kg in the past year was reported and used as a threshold [20]. Unintentional weight loss was obtained by clinical interview at baseline.
BMI	BMI (kg/m^2^) was considered reduced if <20 kg/m^2^ or <22 kg/m^2^ in participants younger and older than 70 years, respectively [19].
Reduced muscle mass	FFMI <17 kg/m² in men and <15 kg/m^2^ in women or ALMI <7 kg/m^2^ in men and <5.5 kg/m^2^ in women was used as a threshold [12,19].
Etiological	
Reduced food intake or assimilation	The first Mini-nutritional Assessment- Short Form (MNA-SF) [21] item was used to determine reduced food intake: “Has food intake declined over the past 3 months due to loss of appetite, digestive problems, chewing or swallowing difficulties?” Severe and moderate decreases were considered positive answers [21]. Chronic gastrointestinal conditions that adversely impact food assimilation or absorption of nutrients were also considered.
Disease burden and inflammation	Interleukin-6 (IL-6) and insulin-like growth factor 1 (IGF-1) were selected as biomarkers to assess inflammation, following recommendations by the Targeting Aging Biomarkers Workgroup for the selection of blood-based biomarkers for geroscience-guided clinical trials [22]. Quartiles for IGF-1 and IL-6 in our own data were calculated in both sexes, and the lowest quartile was considered as a sex-specific threshold: IGF-1 ≤88 ng/mL in men and ≤82 ng/mL in women and IL-6 >3.84 pg/mL in men and >2.99 pg/mL in women [23]. The number of diseases was recorded; disease burden was not assessed.

**Table 2 nutrients-11-02883-t002:** Baseline characteristics of participants in the SarcoPhAge study (*n* = 336).

	Studied Sample (*n* = 336)	Malnutrition According to the ESPEN Criteria			Malnutrition According to the GLIM Criteria		
		Yes (*n* = 19)	No (*n* = 317)	*p*-Value	Yes (*n* = 59)	No (*n* = 277)	*p*-Value
Age, years	72.5 ± 5.8	71.9 ± 7.1	72.6 ± 5.7	0.62	72.0 ± 6.3	72.6 ± 5.7	0.44
Sex, women	186 (55.4%)	13 (68.4%)	173 (54.6%)	0.24	39 (66.1%)	147 (53.1%)	0.07
Number of concomitant diseases per participant	4.1 ± 2.4	5.0 ± 2.8	4.1 ± 2.4	0.09	4.9 ± 2.4	3.9 ± 2.4	0.005
Number of drugs per participant	5.6 ± 3.4	5.9 ± 3.5	5.6 ± 3.4	0.64	6.0 ± 3.3	5.6 ± 3.4	0.29
MMSE, /30 points	28.3 ± 1.8	27.9 ± 1.4	28.3 ± 1.9	0.41	28.0 ± 2.1	28.3 ± 1.7	0.14
Body mass index, kg/m^2^	27.1 ± 4.6	20.9 ± 0.7	27.4 ± 0.2	<0.001	24.0 ± 4.0	27.7 ± 4.5	<0.001
Lean mass total, kg							
Men	56.5 ± 8.7	44.8 ± 4.0	57.0 ± 8.6	0.001	48.7 ± 8.1	57.7 ± 8.2	<0.001
Women	39.0 ± 5.8	35.1 ± 3.5	39.3 ± 5.9	0.013	36.1 ± 4.8	39.7 ± 5.8	0.001
ALMI, kg/m^2^							
Men	8.1 ± 1.0	6.6 ± 0.7	8.1 ± 1.0	0.001	7.1 ± 1.0	8.2 ± 1.0	<0.001
Women	6.1 ± 1.0	5.3 ± 0.5	6.2 ± 1.0	0.003	5.6 ± 0.7	6.3 ± 1.0	<0.001
Muscle strength (kg)							
Men	40.4 ± 8.3	25.8 ± 7.9	41.0 ± 7.7	<0.001	35.7 ± 11.3	41.1 ± 7.5	0.006
Women	22.6 ± 6.9	23.3 ± 6.3	22.5 ± 7.0	0.69	21.7 ± 4.9	22.8 ± 7.3	0.36
Gait speed, m/s	1.02 ± 0.27	1.10 ± 0.30	1.01 ± 0.27	0.21	1.00 ± 0.30	1.03 ± 0.26	0.53
SPPB, /12 points	9.7 ± 2.0	10.2 ± 2.3	9.7 ± 1.9	0.27	9.3 ± 2.5	9.8 ± 1.8	0.08
Chair stand test, s	13.7 ± 5.2	13.6 ± 7.3	13.7 ± 5.0	0.95	14.4 ± 6.1	13.5 ± 5.0	0.26
IADL Lawton							
/5 for men	4.6 ± 1.2	3.8 ± 1.8	4.6 ± 1.2	0.09	4.2 ± 1.7	4.7 ± 1.1	0.08
/8 for women	7.6 ± 1.0	7.3 ± 1.4	7.6 ± 1.0	0.32	7.4 ± 1.3	7.6 ± 0.9	0.24
Level of physical activity, kcal/day	745.7 (270–1523.2)	840 (106–1470)	742 (270–1554)	0.57	935 (150–1470)	735 (270–1568)	0.53

**Table 3 nutrients-11-02883-t003:** Relationship between malnutrition at baseline and the incidence of sarcopenia and severe sarcopenia during a four-year follow-up period (*n* = 336).

**Analysis Performed According to the ESPEN Criteria**					
	**Occurrence of Sarcopenia**		***p*-Value**	**Crude HR (95% CI)**	**Adjusted HR (95% CI) ***
Malnutrition status	No incident sarcopenia (*n* = 290)	Incident sarcopenia (*n* = 46)			
Well nourished	278 (95.9%)	39 (84.8%)	0.005	3.91(1.73–8.81)	4.28 (1.86–9.86)
Malnourished	12 (4.1%)	7 (15.2%)			
	Occurrence of severe sarcopenia		*p*-Value	Crude HR (95% CI)	Adjusted HR (95% CI) *
Malnutrition status	No incident severe sarcopenia (*n* = 310)	Incident severe sarcopenia (*n* = 26)			
Well nourished	295 (95.2%)	22 (84.6%)	0.035	3.54 (1.21–10.34)	3.86 (1.29–11.54)
Malnourished	15 (4.8%)	4 (15.4%)			
**Analysis performed according to the GLIM criteria**					
	Occurrence of sarcopenia		*p*-Value	Crude HR (95% CI)	Adjusted HR (95% CI) *
Malnutrition status	No incident sarcopenia (*n* = 290)	Incident sarcopenia (*n* = 46)			
Well nourished	247 (85.2%)	30 (65.2%)	0.001	3.22 (1.74–5.94)	3.23 (1.73–6.05)
Malnourished	43 (14.8%)	16 (34.8%)			
	Occurrence of severe sarcopenia		*p*-Value	Crude HR (95% CI)	Adjusted HR (95% CI) *
Malnutrition status	No incident severe sarcopenia (*n* = 310)	Incident severe sarcopenia (*n* = 26)			
Well nourished	260 (83.9)	17 (65.4)	0.021	2.90 (1.29–6.53)	2.87 (1.25–6.56)
Malnourished	50 (16.1)	9 (34.6)			

* Covariates: Age, sex, the number of concomitant diseases per participant, the number of drugs per participant, cognitive status, the level of physical activity.

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
