# Peer review of "Malnutrition as a Strong Predictor of the Onset of Sarcopenia"

_nutrients, 2019, doi:10.3390/nu11122883_

Round 1

Reviewer 1 Report

This is an important and very interesting manuscript attempting to describe the relationship between malnutrition and sarcopenia. The study is well conceived- my major concerns are regarding sample size, covariate selection and analysis, and high incidence sarcopenia in non-malnourished participants that may be indicative of poorly defined 'malnutrition'. I also have questions about new onset malnutrition during the study duration, and whether that may have skewed results. Thank you all for your contribution!

Notes:

The introduction may benefit from outlining criteria for malnutrition and sarcopenia (definition), and differences between major entities and their definitions.

Objective: is this a secondary analysis? Please define in objectives.

Line 49: remove those

Lines 50-56: Very long sentence, consider revision

Line 144: How did researchers decide on particular covariates?

Lines 184-186: please provide citation for known covariates

Table 2: are the sample sizes adequate in the ESPEN/GLIM malnutrition groups to assess differences in sarcopenia over time? 19 and 59 are very small samples.

Line 210: Given differences in diagnosis given different definitions, what do authors suggest? (may be discussion point)

Table 3: While there are significant differences among well nourished and malnourished participants in sarcopenia incidence, how do authors explain the overwhelming proportion of sarcopenia, which are actually ‘well nourished’ individuals? This may bear discussion.

Figure 2 is somewhat difficult to see- may consider removing gridlines and making thicker line width, color changes

Discussion:

It may be helpful to re-state objectives in discussion introduction

Was there any analysis of change in malnutrition over the study? It may be that baseline characteristics are unimportant compared to new-onset malnutrition during the study in predicting sarcopenia

Line 306: In covariate analysis, what were key factors in predicting sarcopenia onset outside of malnutrition? Did the covariates adequately cover for factors between groups?

Author Response

REVIEWER 1

Thank you for your interest and the time invested in our manuscript. We really appreciate the opportunity to improve our work and respond to the concerns pointed out in the review.

REVIEWER COMMENT: The introduction may benefit from outlining criteria for malnutrition and sarcopenia (definition), and differences between major entities and their definitions.

AUTHORS’ RESPONSE: We agree that the manuscript will improve if we introduce the definitions from the beginning of the manuscript. These differences have been noted in the Introduction section, as follows:

Introduction (page 3, line 53): “… tree of nutritional disorders. The ESPEN approach is a 2-tier process: in the first step, patients are identified as being at risk of malnutrition by any validated screening tool; in the second step, malnutrition is defined by a combination of weight loss, low body mass index, and low muscle mass (1)”.

Page 4, line 66-77: “… previous definition. The GLIM criteria are a 3-step approach, first, patients are identified by any validated screening tool, and second, they are diagnosed in presence of, at least, one phenotypic (weight loss, low body mass index, and low muscle mass) and one etiologic criteria (reduced food intake or assimilation or disease burden and inflammation). A third step is severity grading, which is based on the phenotypic criteria. The EWGSOP2 consensus follows this same 3-step approach, first, screening by SARC-F questionnaire, and second, patients are diagnosed in presence of low muscle strength and low muscle mass. The third step is severity grading, based on the impairment of physical performance. The GLIM and EWGSOP2 criteria are harmonized definitions that share muscle mass as a criterion to enhance the comparability of studies (3), and sarcopenia has loss of muscle function as its most highlighted differential feature (4)”.

REVIEWER COMMENT: is this a secondary analysis? Please define in objectives.

AUTHORS’ RESPONSE: We understood the comment reviewer but we do not think this study could be considered as a “secondary” analysis of a published study. Indeed, the SarcoPhAge study has not been developed for this specific analysis but this paper is the result of one of all the analyses that this cohort made possible. Therefore, to avoid any confusion, we did not change anything regarding this comment in the manuscript since we already explained that the data from the SarcoPhAge study were used to perform those analyses. Thank you for giving us the opportunity to clarify this important point of our work.

REVIEWER COMMENT: Line 49: remove those

AUTHORS’ RESPONSE: Done, thank you.

REVIEWER COMMENT: Lines 50-56: Very long sentence, consider revision

AUTHORS’ RESPONSE: Thank you. We have modified the paragraph following the reviewer’s suggestion.

Introduction (page 3, lines 47-53): “The European Society of Clinical Nutrition and Metabolism (ESPEN) followed the WHO’s strategy, revisited the concepts of malnutrition and nutrition-related diseases. ESPEN developed malnutrition criteria (1) and guidelines on the definition and terminology of clinical nutrition (2) which unified the terminology to be used in malnutrition and nutrition-related diseases, i.e., sarcopenia, frailty, cachexia/disease-related malnutrition, and starvation-related underweight (1) and organized them as a conceptual tree of nutritional disorders (2).”

REVIEWER COMMENT: Line 144: How did researchers decide on particular covariates?

AUTHORS’ RESPONSE: Many thanks for this comment. We used the covariates collected for the SarcoPhAge study for our analyses. Among all the covariates collected, we used some ‘classical ones’ (age, sex, number of drugs and number of comorbidities) and we also tried to include specific variables that could affect sarcopenia or malnutrition (cognitive status, functional limitations, physical activity). Of course, we could not exclude that some of important covariates have been missed. This interesting point raised by the reviewer has been highlighted in the discussion section.

Discussion section (page 6, lines 401-404): “…, the fact that we used an existing dataset for these analyses implies, first, that we could have miss some important covariates that could explain the incidence of sarcopenia outside of malnutrition and, second, that no power size has been calculated.”

REVIEWER COMMENT: Lines 184-186: please provide citation for known covariates.

AUTHORS’ RESPONSE: Many thanks, some references have been added. We also have modified the sentence with: “... covariates “that could potentially impact” on muscle health and nutritional status … “.

REVIEWER COMMENT: Table 2: are the sample sizes adequate in the ESPEN/GLIM malnutrition groups to assess differences in sarcopenia over time? 19 and 59 are very small samples.

AUTHORS’ RESPONSE: This remark is very interesting, too. We did not performed a statistical power calculation for this specific study since we used data collected from the SarcoPhAge study and included as many subjects as possible. However, such a small sample size already allows the observation of a statistically difference. We can then reasonably assume that the prevalence in a large sample size will be even higher and then results in an even more important difference. We reported this comment in the discussion section.

Discussion (page 6, lines 405-409): However, even with a low prevalence of malnutrition in our sample (n=19 for ESPEN, n=59 for GLIM), a significant relationship with the incidence of sarcopenia has already been found. We can then reasonability assume that a larger sample size and a larger number of malnourished individuals will results in an even more important difference.

REVIEWER COMMENT: Line 210: Given differences in diagnosis given different definitions, what do authors suggest? (may be discussion point)

AUTHORS’ RESPONSE: The reviewer is right, using one definition or another led to a different prevalence of malnutrition. Since the diagnosis criteria are different, it is not surprising to observe a difference in the measured prevalence of malnutrition. However, our purpose was not to define or discuss about a better definition for malnutrition but to highlight that, whatever the diagnosis criteria used, malnutrition is predictive of sarcopenia onset and deserves attention, in order to be diagnosed and treated.

REVIEWER COMMENT: Table 3: While there are significant differences among well-nourished and malnourished participants in sarcopenia incidence, how do authors explain the overwhelming proportion of sarcopenia, which are actually ‘well nourished’ individuals? This may bear discussion.

AUTHORS’ RESPONSE: The reviewer is right. However, since we do not have a dynamic measure of malnutrition, we could not measure if these participants did not change their nutritional status from well-nourished at baseline to malnourished during the follow-up period. Nevertheless, these results highlighted that if participants had a status of malnutrition at baseline, they are at “higher” risk of becoming sarcopenic across a 4-year follow-up period. Malnutrition is therefore considered here as a risk factor of sarcopenia but not the only cause of developing sarcopenia. Of course, this risk factor does not have 100% sensitivity and 100% specificity. Therefore it is understandable to have subjects that are malnourished at baseline and that never develops sarcopenia or, at the opposite, to have subjects well-nourished that will develop sarcopenia. Reviewer’s comment has been included in the discussion section, as follows:

Discussion section (page 6, lines 390-393): “… the dynamic aspect of malnutrition could also be considered with an analysis that does not only focus on the baseline prevalence of malnutrition but that also take into account the incidence or the new-onset of malnutrition as a risk factor for sarcopenia. Unfortunately, in the present study, we did not have the sufficient materials to measure the malnutrition according to both criteria at each time of data collection”.

REVIEWER COMMENT: Figure 2 is somewhat difficult to see- may consider removing gridlines and making thicker line width, color changes

AUTHORS’ RESPONSE: Thank you for this suggestion. We re-designed the figure to provide better visibility.

REVIEWER COMMENT: It may be helpful to re-state objectives in discussion introduction

AUTHORS’ RESPONSE: Of course. Change has been done.

REVIEWER COMMENT: Was there any analysis of change in malnutrition over the study? It may be that baseline characteristics are unimportant compared to new-onset malnutrition during the study in predicting sarcopenia

AUTHORS’ RESPONSE: This suggestion is very relevant. However, due to financial limitations, we could not measure the incidence of new cases of malnutrition over time. Indeed, the diagnostic criteria require the measures of blood parameters and we could not find enough funding for this purpose. We are currently requesting some grants in order to be able to achieve these analyses.  This point has been highlighted in the discussion section.

Discussion section (page 6, lines 390-393): “… the dynamic aspect of malnutrition could also be considered with an analysis that does not only focus on the baseline prevalence of malnutrition but that also take into account the incidence or the new-onset of malnutrition as a risk factor for sarcopenia. Unfortunately, in the present study, we did not have the sufficient materials to measure the malnutrition according to both criteria at each time of data collection”.

REVIEWER COMMENT: Line 306: In covariate analysis, what were key factors in predicting sarcopenia onset outside of malnutrition? Did the covariates adequately cover for factors between groups?

AUTHORS’ RESPONSE: Currently, researchers focused on predictive factors of sarcopenia are still quite poor. Indeed, in the sarcopenia research field, longitudinal prospective studies aiming to measure the consequences of sarcopenia are more usually performed that retrospective studies aiming to measure risk factors for sarcopenia. This is one of the reasons why we decided to perform the current analyses. Therefore, of course, we could have miss important factors in our covariates selection. This has also been recognized as a limitation of our research.

Discussion section (page 6, lines 402-409): Finally, the fact that we used an existing dataset for these analyses implies, first, that we could have miss some important covariates that could explain the incidence of sarcopenia outside of malnutrition and, second, that no power size has been calculated. However, even with a low prevalence of malnutrition in our sample (n=19 for ESPEN, n=59 for GLIM), a significant relationship with the incidence of sarcopenia has already been found. We can then assume that a larger sample size and a larger number of malnourished individuals will results in an even more important difference”.

Reviewer 2 Report

There are few typos and grammatical errors in the whole manuscript. The abstract should be in a single paragraph which very concisely describes the whole work. Here it seems like four different sections. Consider rewriting the abstract in a very clear and easy to understand way. Line number 19, ‘Abstract (200 words max.)’ delete ‘(200 words max.)’ Line number 32, ‘336 had complete data (aged 72.5±5.8; 55.4% women)’ is not clear to me. The exact number of men and women in the data set should be mentioned here. Line number 41, ‘incidence’ does not look as scientific keyword; consider deleting. I could not find the details of any data observed by using the EWGSOP2 criteria.

Author Response

Reviewer 2

REVIEWER COMMENT: There are few typos and grammatical errors in the whole manuscript. The abstract should be in a single paragraph which very concisely describes the whole work. Here it seems like four different sections. Consider rewriting the abstract in a very clear and easy to understand way. Line number 19, ‘Abstract (200 words max.)’ delete ‘(200 words max.)’ Line number 32, ‘336 had complete data (aged 72.5±5.8; 55.4% women)’ is not clear to me. The exact number of men and women in the data set should be mentioned here. Line number 41, ‘incidence’ does not look as scientific keyword; consider deleting. I could not find the details of any data observed by using the EWGSOP2 criteria. 

AUTHORS’ RESPONSE:

Thank you for your interest and the time invested in our manuscript.

Many thanks for your suggestions. The abstract has been rewritten according to your suggestions.

Round 2

Reviewer 1 Report

I am satisfied by the author rebuttals- and recognize the limitations in the current study. Thank you for addressing my questions.